# DISTRIBUTION-BASED INVARIANT DEEP NETWORKS FOR LEARNING META-FEATURES

## ABSTRACT

Recent advances in deep learning from probability distributions successfully achieve classification or regression from distribution samples, thus invariant under permutation of the samples. The first contribution of the paper is to extend these neural architectures to achieve invariance under permutation of the features, too. The proposed architecture, called DIDA, inherits the NN properties of universal approximation, and its robustness with respect to Lipschitz-bounded transformations of the input distribution is established. The second contribution is to empirically and comparatively demonstrate the merits of the approach on two tasks defined at the dataset level. On both tasks, DIDA learns meta-features supporting the characterization of a (labelled) dataset. The first task consists of predicting whether two dataset patches are extracted from the same initial dataset. The second task consists of predicting whether the learning performance achieved by a hyper-parameter configuration under a fixed algorithm (ranging in k-NN, SVM, logistic regression and linear SGD) dominates that of another configuration, for a dataset extracted from the OpenML benchmarking suite. On both tasks, DIDA outperforms the state of the art: DSS and DATASET2VEC architectures, as well as the models based on the hand-crafted meta-features of the literature.

## 1 INTRODUCTION

Deep networks architectures, initially devised for structured data such as images (Krizhevsky et al., 2012) and speech (Hinton et al., 2012), have been extended to enforce some invariance or equivariance properties (Shawe-Taylor, 1993) for more complex data representations. Typically, the network output is required to be invariant with respect to permutations of the input points when dealing with point clouds (Qi et al., 2017), graphs (Henaff et al., 2015) or probability distributions (De Bie et al., 2019). The merit of invariant or equivariant neural architectures is twofold. On the one hand, they inherit the universal approximation properties of neural nets (Cybenko, 1989; Leshno et al., 1993). On the other hand, the fact that these architectures comply with the requirements attached to the data representation yields more robust and more general models, through constraining the neural weights and/or reducing their number.

**Related works.** Invariance or equivariance properties are relevant to a wide range of applications. In the sequence-to-sequence framework, one might want to relax the sequence order (Vinyals et al., 2016). When modelling dynamic cell processes, one might want to follow the cell evolution at a macroscopic level, in terms of distributions as opposed to, a set of individual cell trajectories (Hashimoto et al., 2016). In computer vision, one might want to handle a set of pixels, as opposed to a voxellized representation, for the sake of a better scalability in terms of data dimensionality and computational resources (De Bie et al., 2019).

Neural architectures enforcing invariance or equivariance properties have been pioneered by (Qi et al., 2017; Zaheer et al., 2017) for learning from point clouds subject to permutation invariance or equivariance. These have been extended to permutation equivariance across sets (Hartford et al., 2018). Characterizations of invariance or equivariance under group actions have been proposed in the finite (Gens & Domingos, 2014; Cohen & Welling, 2016; Ravanbakhsh et al., 2017) or infinite case (Wood & Shawe-Taylor, 1996; Kondor & Trivedi, 2018).

On the theoretical side, (Maron et al., 2019a; Keriven & Peyré, 2019) have proposed a general characterization of linear layers enforcing invariance or equivariance properties with respect to the whole permutation group on the feature set. The universal approximation properties of such architectures have been established in the case of sets (Zaheer et al., 2017), point clouds (Qi et al., 2017), equivariant point clouds (Segol & Lipman, 2019), discrete measures (De Bie et al., 2019), invariant (Maron et al., 2019b) and equivariant (Keriven & Peyré, 2019) graph neural networks. The approach most related to our work is that of (Maron et al., 2020), handling point clouds and presenting a neural architecture invariant w.r.t. the ordering of points and their features. In this paper, the proposed *distribution-based invariant deep architecture* (DIDA) extends (Maron et al., 2020) as it handles (discrete or continuous) probability distributions instead of point clouds. This enables to leverage the topology of the Wasserstein distance to provide more general approximation results, covering (Maron et al., 2020) as a special case.

**Motivations.** A main motivation for DIDA is the ability to characterize datasets through *learned meta-features*. Meta-features, aimed to represent a dataset as a vector of characteristics, have been mentioned in the ML literature for over 40 years, in relation with several key ML challenges: (i) learning a performance model, predicting *a priori* the performance of an algorithm (and the hyper-parameters thereof) on a dataset (Rice, 1976; Wolpert, 1996; Hutter et al., 2018); (ii) learning a generic model able of quick adaptation to new tasks, e.g. one-shot or few-shot, through the so-called meta-learning approach (Finn et al., 2018; Yoon et al., 2018); (iii) hyper-parameter transfer learning (Perrone et al., 2018), aimed to transfer the performance model learned for a task, to another task. A large number of meta-features have been manually designed along the years (Muñoz et al., 2018), ranging from sufficient statistics to the so-called *landmarks* (Pfahringer et al., 2000), computing the performance of fast ML algorithms on the considered dataset. Meta-features, expected to describe the joint distribution underlying the dataset, should also be inexpensive to compute. The learning of meta-features has been first tackled by (Jomaa et al., 2019) to our best knowledge, defining the DATASET2VEC representation. Specifically, DATASET2VEC is provided two patches of datasets, (two subsets of examples, described by two (different) sets of features), and is trained to predict whether those patches are extracted from the same initial dataset.

**Contributions.** The proposed DIDA approach extends the state of the art (Maron et al., 2020; Jomaa et al., 2019) in two ways. Firstly, it is designed to handle discrete or continuous probability distributions, as opposed to point sets (Section 2). As said, this extension enables to leverage the more general topology of the Wasserstein distance as opposed to that of the Haussdorf distance (Section 3). This framework is used to derive theoretical guarantees of stability under bounded distribution transformations, as well as universal approximation results, extending (Maron et al., 2020) to the continuous setting. Secondly, the empirical validation of the approach on two tasks defined at the dataset level demonstrates the merit of the approach compared to the state of the art (Maron et al., 2020; Jomaa et al., 2019; Muñoz et al., 2018) (Section 4).

**Notations.** $[\![1; m]\!]$ denotes the set of integers $\{1, \ldots m\}$. Distributions, including discrete distributions (datasets) are noted in bold font. Vectors are noted in italic, with $x[k]$ denoting the $k$-th coordinate of vector $x$.

## 2 DISTRIBUTION-BASED INVARIANT NETWORKS FOR META-FEATURE LEARNING

This section describes the core of the proposed distribution-based invariant neural architectures, specifically the mechanism of mapping a point distribution onto another one subject to sample and feature permutation invariance, referred to as *invariant layer*. For the sake of readability, this section focuses on the case of discrete distributions, referring the reader to Appendix A for the general case of continuous distributions.

### 2.1 INVARIANT FUNCTIONS OF DISCRETE DISTRIBUTIONS

Let $\mathbf{z} = \{(x_i, y_i) \in \mathbb{R}^d, i \in [\![1; n]\!]\}$ denote a dataset including $n$ labelled samples, with $x_i \in \mathbb{R}^{d_X}$ an instance and $y_i \in \mathbb{R}^{d_Y}$ the associated multi-label. With $d_X$ and $d_Y$ respectively the dimensions

of the instance and label spaces, let $d \stackrel{\text{def.}}{=} d_X + d_Y$. By construction, $\mathbf{z}$ is invariant under permutation on the sample ordering; it is viewed as an $n$-size discrete distribution $\frac{1}{n} \sum_{i=1}^{n} \delta_{z_i}$ in $\mathbb{R}^d$ with $\delta_{z_i}$ the Dirac function at $z_i$. In the following, $Z_n(\mathbb{R}^d)$ denotes the space of such $n$-size point distributions, with $Z(\mathbb{R}^d) \stackrel{\text{def.}}{=} \cup_n Z_n(\mathbb{R}^d)$ the space of distributions of arbitrary size.

Let $G \stackrel{\text{def.}}{=} S_{d_X} \times S_{d_Y}$ denote the group of permutations independently operating on the feature and label spaces. For $\sigma = (\sigma_X, \sigma_Y) \in G$, the image $\sigma(z)$ of a labelled sample is defined as $(\sigma_X(x), \sigma_Y(y))$, with $x = (x[k], k \in [\![1; d_X]\!])$ and $\sigma_X(x) \stackrel{\text{def.}}{=} (x[\sigma_X(k)], k \in [\![1; d_X]\!])$. For simplicity and by abuse of notations, the operator mapping a distribution $\mathbf{z} = (z_i, i \in [\![1; n]\!])$ to $\{\sigma(z_i)\} \stackrel{\text{def.}}{=} \sigma_\sharp \mathbf{z}$ is still denoted $\sigma$.

Let $Z(\Omega)$ denote the space of distributions supported on some domain $\Omega \subset \mathbb{R}^d$, with $\Omega$ invariant under permutations in $G$. The goal of the paper is to define and train deep architectures, implementing functions $\varphi$ on $Z(\Omega \subset \mathbb{R}^d)$ that are invariant under $G$, i.e. such that $\forall \sigma \in G, \varphi(\sigma_\sharp \mathbf{z}) = \varphi(\mathbf{z})$[1]. By construction, a multi-label dataset is invariant under permutations of the samples, of the features, and of the multi-labels. Therefore, any meta-feature, that is, a feature describing a multi-label dataset, is required to satisfy the above sample and feature permutation invariance properties.

## 2.2 DISTRIBUTION-BASED INVARIANT LAYERS

The building block of the proposed architecture, the invariant layer meant to satisfy the feature and label invariance requirements, is defined as follows, taking inspiration from (De Bie et al., 2019).

**Definition 1.** *(Distribution-based invariant layers) Let an interaction functional $\varphi : \mathbb{R}^d \times \mathbb{R}^d \to \mathbb{R}^r$ be G-invariant:*

$$\forall \sigma \in G, \quad \forall (z_1, z_2) \in \mathbb{R}^d \times \mathbb{R}^d, \quad \varphi(z_1, z_2) = \varphi(\sigma(z_1), \sigma(z_2)).$$

*The distribution-based invariant layer $f_\varphi$ is defined as*

$$f_\varphi : \mathbf{z} = (z_i)_{i \in [\![1;n]\!]} \in Z(\mathbb{R}^d) \mapsto f_\varphi(\mathbf{z}) \stackrel{\text{def.}}{=} \left( \frac{1}{n} \sum_{j=1}^{n} \varphi(z_1, z_j), \ldots, \frac{1}{n} \sum_{j=1}^{n} \varphi(z_n, z_j) \right) \in Z(\mathbb{R}^r). \tag{1}$$

By construction, $f_\varphi$ is $G$-invariant if $\varphi$ is $G$-invariant. The construction of $f_\varphi$ is extended to the general case of possibly continuous probability distributions by essentially replacing sums by integrals (Appendix A)

*Remark* 1. (Varying dimensions $d_X$ and $d_Y$). Both in practice and in theory, it is important that $f_\varphi$ layers (in particular the first layer of the neural architecture) handle datasets of arbitrary number of features $d_X$ and number of multi-labels $d_Y$. The proposed approach, used in the experiments (Section 4), is to define $\varphi$ as follows. Letting $z = (x, y)$ and $z' = (x', y')$ be two samples in $\mathbb{R}^{d_X} \times \mathbb{R}^{d_Y}$, let $u$ be defined from $\mathbb{R}^4$ onto $\mathbb{R}^t$, consider the sum of $u(x[k], x'[k], y[\ell], y'[\ell])$ for $k$ ranging in $[\![1; d_X]\!]$ and $\ell$ in $[\![1; d_Y]\!]$, and apply mapping $v$ from $\mathbb{R}^t$ to $\mathbb{R}^r$ on this sum:

$$\varphi(z, z') = v \left( \sum_{k=1}^{d_X} \sum_{\ell=1}^{d_Y} u(x[k], x'[k], y[\ell], y'[\ell]) \right) \tag{2}$$

Likewise, by construction $\varphi$ is $G$-invariant, i.e. it is invariant to both feature and label permutations. As shown in Section 4, this invariance property is instrumental to a good empirical performance.

*Remark* 2. (Varying sample size $n$). By construction, $f_\varphi$ is defined on $Z(\mathbb{R}^d) = \cup_n Z_n(\mathbb{R}^d)$ (independent of $n$), such that it supports inputs of arbitrary cardinality $n$.

*Remark* 3. (Discussion w.r.t. (Maron et al., 2020)) The above definition of $f_\varphi$ is based on the aggregation of pairwise terms $\varphi(z_i, z_j)$. The motivation for using a pairwise $\varphi$ is twofold. On the one hand, capturing local sample interactions allows to create more expressive architectures, which is important to improve the performance on some complex data sets, as illustrated in the experiments (Section 4). On the other hand, interaction functionals are crucial to design universal architectures

---

[1]As opposed to *G-equivariant* functions that are characterized by $\forall \sigma \in G, \varphi(\sigma_\sharp \mathbf{z}) = \sigma_\sharp \varphi(\mathbf{z})$

(Appendix C, theorem 2). The proposed theoretical framework relies on the Wasserstein distance (corresponding to the convergence in law of probability distributions), which enables to compare distributions with varying number of points or even with continuous densities. In contrast, Maron et al. (2020) do not use interaction functionals, and establish the universality of their DSS architecture for fixed dimension $d$ and number of points $n$. Moreover, DSS happens to resort to max pooling operators, discontinuous w.r.t. the Wasserstein topology (see Remark 6).

Two particular cases are when $\varphi$ only depends on its first or second input:

(i) if $\varphi(z, z') = \psi(z')$, then $f_\varphi$ computes a global "moment" descriptor of the input, as $f_\varphi(\mathbf{z}) = \frac{1}{n} \sum_{j=1}^{n} \psi(z_j) \in \mathbb{R}^r$.

(ii) if $\varphi(z, z') = \xi(z)$, then $f_\varphi$ transports the input distribution via $\xi$, as $f_\varphi(\mathbf{z}) = \{\xi(z_i), i \in [\![1; n]\!]\} \in Z(\mathbb{R}^r)$. This operation is referred to as a *push-forward*.

*Remark* 4. (Localized computation) In practice, the quadratic complexity of $f_\varphi$ w.r.t. the number $n$ of samples can be reduced by only computing $\varphi(z_i, z_j)$ for pairs $z_i, z_j$ sufficiently close to each other. Layer $f_\varphi$ thus extracts and aggregates information related to the neighborhood of the samples.

*Remark* 5. (Link to kernels) The use of an interaction functional $\varphi$ is inspired from kernel ideas, albeit with significant differences: (i) in $f_\varphi(z_i)$, the detail of the pairwise interactions $\varphi(z_i, z_j)$ is lost through averaging; (ii) $\varphi$ takes into account labels; (iii) $\varphi$ is learnt. Further work will be devoted to investigating the properties of the $f_\varphi(z_i)$ matrix.

### 2.3 LEARNING META-FEATURES

The proposed distributional neural architectures defined on point distributions (DIDA) are sought as

$$\mathbf{z} \in Z(\mathbb{R}^d) \mapsto \mathcal{F}_\zeta(\mathbf{z}) \overset{\text{def.}}{=} f_{\varphi_m} \circ f_{\varphi_{m-1}} \circ \ldots \circ f_{\varphi_1}(\mathbf{z}) \in \mathbb{R}^{d_{m+1}} \tag{3}$$

where $\zeta$ are the trainable parameters of the architecture (below). Only the case $d_Y = 1$ is considered in the remainder. The $k$-th layer is built on the top of $\varphi_k$, mapping pairs of vectors in $\mathbb{R}^{d_k}$ onto $\mathbb{R}^{d_{k+1}}$, with $d_1 = d$ (the dimension of the input samples). Last layer is built on $\varphi_m$, only depending on its second argument; it maps the distribution in layer $m-1$ onto a vector, whose coordinates are referred to as meta-features.

The $G$-invariance and dimension-agnosticity of the whole architecture only depend on the first layer $f_{\varphi_1}$ satisfying these properties. In the first layer, $\varphi_1$ is sought as $\varphi_1((x, y), (x', y')) = v(\sum_k u(x[k], x'[k], y, y'))$ (Remark 1), with $u(x[k], x'[k], y, y') = (\rho(A_u \cdot (x[k]; x'[k]) + b_u, \mathbb{1}_{y \neq y'})$ in $\mathbb{R}^t \times \{0, 1\}$, where $\rho$ is a non-linear activation function, $A_u$ a $(t, 2)$ matrix, $(x[k]; x'[k])$ the 2-dimensional vector concatenating $x[k]$ and $x'[k]$, and $b_u$ a $t$-dimensional vector. With $e = \sum_k u(x[k], x'[k], y, y')$, function $v$ likewise applies a non-linear activation function $\rho$ on an affine transformation of $e$: $v(e) = \rho(A_v \cdot e + b_v)$, with $A_v$ a $(t, r)$ matrix and $b_v$ a $r$-dimensional vector.

Note that the subsequent layers need neither be invariant w.r.t. the number of samples, nor handle a varying number of dimensions. However, maintaining the distributional nature among several layers is shown to improve performance in practice (Section 4). Every $\varphi_k, k \geq 2$ is defined as $\varphi_k = \rho(A_k \cdot + b_k)$, with $\rho$ an activation function, $A_k$ a $(d_k, d_{k+1})$ matrix and $b_k$ a $d_{k+1}$-dimensional vector. The DIDA neural net thus is parameterized by $\zeta \overset{\text{def.}}{=} (A_u, b_u, A_v, b_v, \{A_k, b_k\}_k)$, that is classically learned by stochastic gradient descent from the loss function defined after the task at hand (Section 4).

## 3 THEORETICAL ANALYSIS

This section analyzes the properties of invariant-layer based neural architectures, specifically their robustness w.r.t. bounded transformations of the involved distributions, and their approximation abilities w.r.t. the convergence in law, which is the natural topology for distributions. As already said, the discrete distribution case is considered in this section for the sake of readability, referring the reader to Appendix A for the general case of continuous distributions.

### 3.1 OPTIMAL TRANSPORT COMPARISON OF DATASETS

**Point clouds vs. distributions.** Our claim is that datasets should be seen as probability distributions, rather than point clouds. Typically, including many copies of a point in a dataset amounts to increasing its importance, which usually makes a difference in a standard machine learning setting. Accordingly, the topological framework used to define and learn meta-features in the following is that of the convergence in law, with the distance among two datasets being quantified using the Wasserstein distance (below). In contrast, the point clouds setting (see for instance (Qi et al., 2017)) relies on the Haussdorff distance among sets to theoretically assess the robustness of these architectures. While it is standard for 2D and 3D data involved in graphics and vision domains, it faces some limitations in higher dimensional domains, e.g. due to max-pooling being a non-continuous operator w.r.t. the convergence in law topology.

**Wasserstein distance.** Referring the reader to (Santambrogio, 2015; Peyré & Cuturi, 2019) for a more comprehensive presentation, the standard 1-Wasserstein distance between two discrete probability distributions $\mathbf{z}, \mathbf{z}' \in Z_n(\mathbb{R}^d) \times Z_m(\mathbb{R}^d)$ is defined as:

$$W_1(\mathbf{z}, \mathbf{z}') \stackrel{\text{def.}}{=} \max_{f \in \text{Lip}_1(\mathbb{R}^d)} \frac{1}{n} \sum_{i=1}^{n} f(z_i) - \frac{1}{m} \sum_{j=1}^{m} f(z'_j)$$

with $\text{Lip}_1(\mathbb{R}^d)$ the space of 1-Lipschitz functions $f : \mathbb{R}^d \to \mathbb{R}$. To account for the invariance requirement (making indistinguishable $\mathbf{z} = (z_1, \ldots, z_n)$ and its permuted image $(\sigma(z_1), \ldots, \sigma(z_n)) \stackrel{\text{def.}}{=} \sigma_\sharp \mathbf{z}$ under $\sigma \in G$), we introduce the $G$-invariant 1-Wasserstein distance: for $\mathbf{z} \in Z_n(\mathbb{R}^d), \mathbf{z}' \in Z_m(\mathbb{R}^d)$:

$$\overline{W}_1(\mathbf{z}, \mathbf{z}') = \min_{\sigma \in G} W_1(\sigma_\sharp \mathbf{z}, \mathbf{z}')$$

such that $\overline{W}_1(\mathbf{z}, \mathbf{z}') = 0$ if and only if $\mathbf{z}$ and $\mathbf{z}'$ belong to the same equivalence class (Appendix A), i.e. are equal in the sense of probability distributions up to sample and feature permutations.

**Lipschitz property.** In this context, a map $f$ from $Z(\mathbb{R}^d)$ onto $Z(\mathbb{R}^r)$ is continuous for the convergence in law (a.k.a. weak convergence on distributions, denoted $\rightharpoonup$) iff for any sequence $\mathbf{z}^{(k)} \rightharpoonup \mathbf{z}$, then $f(\mathbf{z}^{(k)}) \rightharpoonup f(\mathbf{z})$. The Wasserstein distance metrizes the convergence in law, in the sense that $\mathbf{z}^{(k)} \rightharpoonup \mathbf{z}$ is equivalent to $W_1(\mathbf{z}^{(k)}, \mathbf{z}) \to 0$. Furthermore, map $f$ is said to be $C$-Lipschitz for the permutation invariant 1-Wasserstein distance iff

$$\forall \mathbf{z}, \mathbf{z}' \in Z(\mathbb{R}^d), \quad \overline{W}_1(f(\mathbf{z}), f(\mathbf{z}')) \leqslant C \overline{W}_1(\mathbf{z}, \mathbf{z}'). \tag{4}$$

The $C$-Lipschitz property entails the continuity of $f$ w.r.t. its input: if two input distributions are close in the permutation invariant 1-Wasserstein sense, the corresponding outputs are close too.

### 3.2 REGULARITY OF DISTRIBUTION-BASED INVARIANT LAYERS

Assuming the interaction functional to satisfy the Lipschitz property:

$$\forall z \in \mathbb{R}^d, \quad \varphi(z, \cdot) \quad \text{and} \quad \varphi(\cdot, z) \quad \text{are} \quad C_\varphi - \text{Lipschitz.} \tag{5}$$

the robustness of invariant layers with respect to different variations of their input is established (proofs in Appendix B). We first show that invariant layers also satisfy Lipschitz property, ensuring that deep architectures of the form (3) map close inputs onto close outputs.

**Proposition 1.** *Invariant layer $f_\varphi$ of type (1) is $(2rC_\varphi)$-Lipschitz in the sense of (4).*

A second result regards the case where two datasets $\mathbf{z}$ and $\mathbf{z}'$ are such that $\mathbf{z}'$ is the image of $\mathbf{z}$ through some diffeomorphism $\tau$ ($\mathbf{z} = (z_1, \ldots, z_n)$ and $\mathbf{z}' = \tau_\sharp \mathbf{z} = (\tau(z_1), \ldots, \tau(z_n))$. If $\tau$ is close to identity, then the following proposition shows that $f_\varphi(\tau_\sharp \mathbf{z})$ and $f_\varphi(\mathbf{z})$ are close too. More generally, if continuous transformations $\tau$ and $\xi$ respectively apply on the input and output space of $f_\varphi$, and are close to identity, then $\xi_\sharp f_\varphi(\tau_\sharp \mathbf{z})$ and $f_\varphi(\mathbf{z})$ are also close.

**Proposition 2.** *Let $\tau : \mathbb{R}^d \to \mathbb{R}^d$ and $\xi : \mathbb{R}^r \to \mathbb{R}^r$ be two Lipschitz maps with respectively Lipschitz constants $C_\tau$ and $C_\xi$. Then,*

$$\forall \mathbf{z} \in Z(\Omega), \ \overline{W}_1(\xi_\sharp f_\varphi(\tau_\sharp \mathbf{z}), f_\varphi(\mathbf{z})) \leqslant \sup_{x \in f_\varphi(\tau(\Omega))} \|\xi(x) - x\|_2 + 2r \, \text{Lip}(\varphi) \sup_{x \in \Omega} \|\tau(x) - x\|_2$$

$$\forall \mathbf{z}, \mathbf{z}' \in Z(\Omega), \ \text{if } \tau \text{ is equivariant,} \ \overline{W}_1(\xi_\sharp f_\varphi(\tau_\sharp \mathbf{z}), \xi_\sharp f_\varphi(\tau_\sharp \mathbf{z}')) \leqslant 2r \, C_\varphi \, C_\tau \, C_\xi \overline{W}_1(\mathbf{z}, \mathbf{z}')$$

### 3.3 Universality of Invariant Layers

Lastly, the universality of the proposed architecture is established, showing that the composition of an invariant layer (1) and a fully-connected layer is enough to enjoy the universal approximation property, over all functions defined on $Z(\mathbb{R}^d)$ with dimension $d$ less than some $D$ (Remark 1).

**Theorem 1.** *Let $\mathcal{F} : Z(\Omega) \to \mathbb{R}$ be a $G$-invariant map on a compact $\Omega$, continuous for the convergence in law. Then $\forall \varepsilon > 0$, there exists two continuous maps $\psi, \varphi$ such that*

$$\forall \mathbf{z} \in Z(\Omega), \quad |\mathcal{F}(\mathbf{z}) - \psi \circ f_\varphi(\mathbf{z})| < \varepsilon$$

*where $\varphi$ is $G$-invariant and independent of $\mathcal{F}$.*

*Proof.* The sketch of the proof is as follows (complete proof in Appendix C). Let us define $\varphi = g \circ h$ where: (i) $h$ is the collection of $d_X$ elementary symmetric polynomials in the features and $d_Y$ elementary symmetric polynomials in the labels, which is invariant under $G$; (ii) a discretization of $h(\Omega)$ on a grid is then considered, achieved thanks to $g$ that aims at collecting integrals over each cell of the discretization; (iii) $\psi$ applies function $\mathcal{F}$ on this discretized measure; this requires $h$ to be bijective, and is achieved by $\tilde{h}$, through a projection on the quotient space $S_d/G$ and a restriction to its image compact $\Omega'$. To sum up, $f_\varphi$ defined as such computes an expectation which collects integrals over each cell of the grid to approximate measure $h_\sharp \mathbf{z}$ by a discrete counterpart $\widehat{h_\sharp \mathbf{z}}$. Hence $\psi$ applies $\mathcal{F}$ to $\tilde{h}_\sharp^{-1}(\widehat{h_\sharp \mathbf{z}})$. Continuity is obtained as follows: (i) proximity of $h_\sharp \mathbf{z}$ and $\widehat{h_\sharp \mathbf{z}}$ follows from Lemma 1 in (De Bie et al., 2019)) and gets tighter as the grid discretization step tends to 0; (ii) Map $\tilde{h}^{-1}$ is $1/d$-Hölder, after Theorem 1.3.1 from (Rahman & Schmeisser, 2002)); therefore Lemma 2 entails that $\overline{W}_1(\mathbf{z}, \tilde{h}_\sharp^{-1} \widehat{h_\sharp \mathbf{z}})$ can be upper-bounded; (iii) since $\Omega$ is compact, by Banach-Alaoglu theorem, $Z(\Omega)$ also is. Since $\mathcal{F}$ is continuous, it is thus uniformly weakly continuous: choosing a discretization step small enough ensures the result. $\square$

*Remark* 6. (Comparison with (Maron et al., 2020)) The above proof holds for functionals of arbitrary input sample size $n$, as well as continuous distributions, generalizing results in (Maron et al., 2020). Note that the two types of architectures radically differ (more in Section 4).

*Remark* 7. (Approximation by an invariant NN) After theorem 1, any invariant continuous function defined on distributions with compact support can be approximated with arbitrary precision by an invariant neural network (Appendix C). The proof involves mainly three steps: (i) an invariant layer $f_\varphi$ can be approximated by an invariant network; (ii) the universal approximation theorem (Cybenko, 1989; Leshno et al., 1993); (iii) uniform continuity is used to obtain uniform bounds.

*Remark* 8. (Extension to different spaces) Theorem 1 also extends to distributions supported on different spaces, via embedding them into a high-dimensional space. Therefore, any invariant function on distributions with compact support in $\mathbb{R}^d$ with $d \leq D$ can be uniformly approximated by an invariant network (Appendix C).

## 4 Experimental validation

The experimental validation presented in this section considers two goals of experiments: (i) assessing the ability of DIDA to learn accurate meta-features; (ii) assessing the merit of the DIDA invariant layer design, building invariant $f_\varphi$ on the top of an interactional function $\varphi$ (Eq. 1). As said, this architecture is expected to grasp contrasts among samples, e.g. belonging to different classes; the proposed experimental setting aims to empirically investigate this conjecture.

**Baselines.** These goals of experiments are tackled by comparing DIDA to three baselines: DSS layers (Maron et al., 2020); hand-crafted meta-features (HC) (Muñoz et al., 2018) (Table 5 in Appendix D); DATASET2VEC (Jomaa et al., 2019). We implemented DSS, the code being not available.[2] In order to cope with varying dataset dimensions (as required by the UCI and OpenML benchmarks), the original DSS was augmented with an aggregator summing over the features. Three DSS baselines are considered: linear or non-linear invariant layers, possibly preceded by equivariant layers. Similarly, the original DATASET2VEC implementation has been augmented to address our experimental setting. The baselines are detailed in Appendix D.3.

---

[2]The code source of DIDA and (our implementation of) baselines are available in supplementary materials.

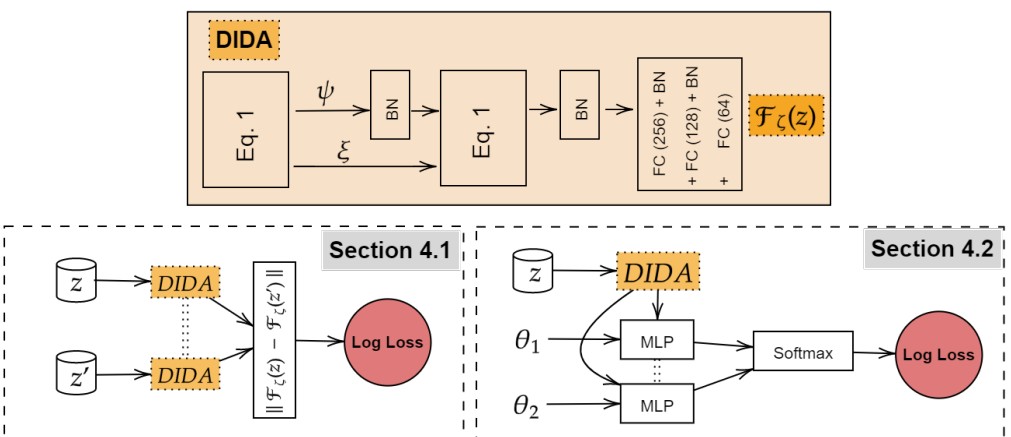

Figure 1: Learning meta-features with DIDA. Top: the DIDA architecture (BN stands for batch norm; FC for fully connected layer). Bottom left: Learning meta-features for patch identification using a Siamese architecture (section 4.1). Bottom right: learning meta-features for performance modelling, specifically to rank two hyper-parameter configurations $\theta_1$ and $\theta_2$ (section 4.2).

**Experimental setting.** Two tasks defined at the dataset level are considered: patch identification (section 4.1) and performance modelling (section 4.2). The dataset preprocessing protocols are detailed in Appendix D.1. On both tasks, the same DIDA architecture is considered (Fig 1), involving 2 invariant layers followed by 3 fully connected (FC) layers. Meta-features $\mathcal{F}_\zeta(\mathbf{z})$ consist of the output of the third FC layer, with $\zeta$ denoting the trained DIDA () parameters. All experiments run on 1 NVIDIA-Tesla-V100-SXM2 GPU with 32GB memory, using Adam optimizer with base learning rate $10^{-3}$.

### 4.1 TASK 1: PATCH IDENTIFICATION

The patch identification task consists of detecting whether two blocks of data are extracted from the same original dataset (Jomaa et al., 2019). Letting $\mathbf{u}$ denote a $n$-sample, $d$-dimensional dataset, an $n_z, d_z$ patch $\mathbf{z}$ is constructed from $\mathbf{u}$ by selecting $n_z$ examples in $\mathbf{u}$ (sampled uniformly with replacement) and retaining their description along $d_z$ features (sampled uniformly with replacement). The size $n_z$ and number $d_z$ of features of the patch are uniformly selected in fixed intervals (Table 4, Appendix D). To each pair of patches $\mathbf{z}, \mathbf{z}'$ with same number of instances $n_z = n_{z'}$, is associated a binary meta-label $\ell(\mathbf{z}, \mathbf{z}')$ set to 1 iff $\mathbf{z}$ and $\mathbf{z}$' are extracted from the same initial dataset $\mathbf{u}$. DIDA parameters $\zeta$ are trained to minimize the cross-entropy loss of model $\hat{\ell}_\zeta(\mathbf{z}, \mathbf{z}') = \exp\left(-\|\mathcal{F}_\zeta(\mathbf{z}) - \mathcal{F}_\zeta(\mathbf{z}')\|_2\right)$, with $\mathcal{F}_\zeta(\mathbf{z})$ and $\mathcal{F}_\zeta(\mathbf{z}')$ the meta-features computed for $\mathbf{z}$ and $\mathbf{z}'$:

$$\text{Minimize } \mathcal{L}(\zeta) = -\sum_{\mathbf{z}, \mathbf{z}'} \ell(\mathbf{z}, \mathbf{z}') \log(\hat{\ell}_\zeta(\mathbf{z}, \mathbf{z}')) + (1 - \ell(\mathbf{z}, \mathbf{z}')) \log(1 - \hat{\ell}_\zeta(\mathbf{z}, \mathbf{z}')) \quad (6)$$

DIDA and all baselines are trained using a Siamese approach (Figure 1, bottom left): the same DIDA (or baseline) architecture is used to compute meta-features $\mathcal{F}_\zeta(\mathbf{z})$ and $\mathcal{F}_\zeta(\mathbf{z}')$ from patches $\mathbf{z}$ and $\mathbf{z}'$, and trained to minimize the cross-entropy loss w.r.t. $\ell(\mathbf{z}, \mathbf{z}')$. The classification results on toy datasets and UCI datasets (Table 1, detailed in Appendix D) show the pertinence of the DIDA meta-features, particularly so on the UCI datasets where the number of features widely varies from one dataset to another. The relevance of the interactional invariant layer design is established on this problem as DIDA outperforms both DATASET2VEC, DSS as well as the function learned on the top of the hand-crafted meta-features.

**An ablation study** is conducted to assess the impact of (i) the feature permutation invariance; (ii) considering one *vs* two invariant layers of type (1). The so-called NO-FINV-DSS baseline, detailed in Appendix D, is built upon (Zaheer et al., 2017); it only differs from the DSS baseline as it is *not* feature permutation invariant. With ca the same number of parameters as DSS, its performances are

| Method | # parameters | TOY | UCI |
|---|---|---|---|
| Hand-crafted | 53312 | 77.05 %± 1.63 | 58.36 %± 2.64 |
| NO-FINV-DSS (no invariance in features) | 1297692 | 90.49 %± 1.73 | 64.69 %± 4.89 |
| DATASET2VEC (our implementation) | 257088 | 97.90 %± 1.87 | 77.0.5 %± 3.49 |
| DSS layers (Linear aggregation) | 1338684 | 89.32 %± 1.85 | 76.23 %± 1.84 |
| DSS layers (Non-linear aggregation) | 1338684 | 96.24 %± 2.04 | 83.97 %± 2.89 |
| DSS layers (Equivariant+invariant) | 1338692 | 96.26 %± 1.40 | 82.94 %± 3.36 |
| DIDA (1 invariant layer) | 323028 | 91.37 %± 1.39 | 81.03 %± 3.23 |
| DIDA (2 invariant layers) | 1389089 | 97.20 % ± 0.10 | **89.70** % ± **1.89** |

Table 1: Patch identification (binary classification accuracy) on 10 runs of DIDA and considered baselines.

| Method | SGD | SVM | LR | k-NN |
|---|---|---|---|---|
| Hand-crafted | 71.18 %± 0.41 | 75.39 %± 0.29 | 86.41 %± 0.419 | 65.44 %± 0.73 |
| DATASET2VEC | 74.43 %± 0.90 | 81.75 %± 1.85 | 89.18 %± 0.45 | 72.90 %± 1.13 |
| DSS (Linear aggregation) | 73.46 %± 1.44 | 82.91 %± 0.22 | 87.93 %± 0.58 | 70.07 %± 2.82 |
| DSS (Equivariant+Invariant) | 73.54 %± 0.26 | 81.29 %± 1.65 | 87.65 %± 0.03 | 68.55 %± 2.84 |
| DSS (Non-linear aggregation) | 74.13 %± 1.01 | 83.38 %± 0.37 | 87.92 %± 0.27 | 73.07 %± 0.77 |
| DIDA (1 invariant layer) | 77.31 %± 0.16 | 84.05 %± 0.71 | **90.16** %± **0.17** | 74.41 %± 0.93 |
| DIDA (2 invariant layers) | **78.41** %± **0.41** | **84.14** %± **0.02** | 89.77 %± 0.50 | **78.91** %± **0.54** |

Table 2: Pairwise ranking of configurations, for ML algorithms SGD, SVM, LR and k-NN: performance on test set of DIDA, hand-crafted, DATASET2VEC and DSS (average and std deviation on 3 runs).

significantly lower (Table 1), showcasing the benefits of enforcing the feature invariance property. Secondly, we compare the 2-invariant layers DIDA, with the 1-invariant layer DIDA (1L-DIDA and 2L-DIDA for short): 1L-DIDA yields significantly lower performances, which confirms the advantages of maintaining the distributional nature among several layers, as already noted by (De Bie et al., 2019). Note that the 1L-DIDA still outperforms the non feature-invariant baseline, while requiring much fewer parameters.

## 4.2 TASK 2: PERFORMANCE MODEL LEARNING

The performance modelling task aims to assess *a priori* the accuracy of the classifier learned from a given machine learning algorithm with a given configuration $\theta$ (vector of hyper-parameters ranging in a hyper-parameter space $\Theta$, Table 6 in Appendix D), on a dataset $\mathbf{z}$ (for brevity, the performance of $\theta$ on $\mathbf{z}$) (Rice, 1976).

For each ML algorithm, ranging in Logistic regression (LR), SVM, k-Nearest Neighbours (k-NN), linear classifier learned with stochastic gradient descent (SGD), a set of meta-features is learned to predict whether some configuration $\theta_1$ outperforms some configuration $\theta_2$ on dataset $\mathbf{z}$: to each triplet $(\mathbf{z}, \theta_1, \theta_2)$ is associated a binary value $\ell(\mathbf{z}, \theta_1, \theta_2)$, set to 1 iff $\theta_2$ yields better performance than $\theta_1$ on $\mathbf{z}$. DIDA parameters $\zeta$ are trained to build model $\hat{\ell}_\zeta$, minimizing the (weighted version of) cross-entropy loss (6), where $\hat{\ell}_\zeta(\mathbf{z}, \theta_1, \theta_2)$ is a 2-layer FC network with input vector $[\mathcal{F}_\zeta(\mathbf{z}); \theta_1; \theta_2]$, depending on the considered ML algorithm and its configuration space.

In each epoch, a batch made of triplets $(\mathbf{z}, \theta_1, \theta_2)$ is built, with $\theta_1, \theta_2$ uniformly drawn in the algorithm configuration space (Table 6) and $\mathbf{z}$ a $n$-sample $d$-dimensional patch of a dataset in the OpenML CC-2018 (Bischl et al., 2019) with $n$ uniformly drawn in $[700; 900]$ and $d$ in $[3; 10]$. Algorithm 1 summarizes the training procedure.

The quality of the DIDA meta-features is assessed from the ranking accuracy (Table 2), showing their relevance. The performance gap compared to the baselines is higher for the k-NN modelling

---

**Algorithm 1** Performance Modeling

---

1: $\mathcal{F}_\zeta \leftarrow$ meta-feature extractor (DIDA, DSS, DATASET2VEC, or Hand-crafted)
2: MLP $\leftarrow$ NN[Linear(64)-ReLU-Linear(32)-ReLU-Linear(1)]
3: CLF $\leftarrow$ machine learning classifier (SGD, SVM, LR or k-NN)
4: error $\leftarrow$ 3-CV classification error function
5: **for** iteration=$1, 2, \ldots$ **do**
6:     Sample $(\theta_1, \theta_2)$, two hyper-parameters of CLF          $\triangleright$ Search space: Table 6
7:     Sample patch $\mathbf{z}$ from dataset $u$          $\triangleright$ Patch dimension: Table 4
8:     pred $\leftarrow$ softmax(MLP($\mathcal{F}_\zeta(\mathbf{z}), \theta_1$), MLP($\mathcal{F}_\zeta(\mathbf{z}), \theta_2$))
9:     Backpropagate logloss(pred, 0 if error($\mathbf{z}$, CLF($\theta_1$)) < error($\mathbf{z}$, CLF($\theta_2$)) else 1)
10: **end for**

---

task; this is explained as the sought performance model only depends on the local geometry of the examples. Still, good performances are observed over all considered algorithms. Note that the 2L-DIDA yields significantly better (respectively, similar) performances than 1L-DIDA on the $k$-NN model (resp. on all other models).

**Meta-feature assessment.** A regression setting is thereafter considered, aimed to predict the actual performance of a configuration $\theta$ based on the (frozen) meta-features $\mathcal{F}_\zeta(\mathbf{z})$. The regression accuracy is illustrated for the configurations of the $k$-NN algorithm on Figure 2, left (results for other algorithms are presented in Appendix D). The comparison with the regression models based on DSS meta-features or hand-crafted features (Figure 2, middle and right) shows the merits of the DIDA architecture; a tentative interpretation for the DIDA better performance is based on the interactional nature of DIDA architecture, better capturing local interactions.

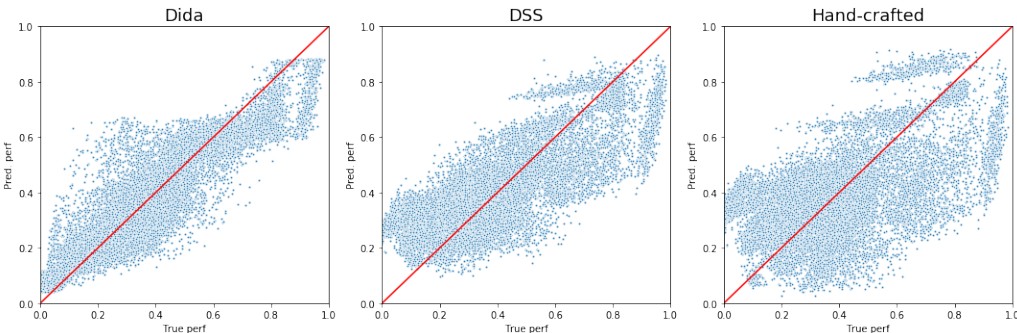

Figure 2: k-NN: True performance vs performance predicted by regression on top of the meta-features (i) learned by DIDA, (ii) DSS or (iii) Hand-crafted statistics.

## 5  CONCLUSION

The theoretical contribution of the paper is the DIDA architecture, able to learn from discrete and continuous distributions on $\mathbb{R}^d$, invariant w.r.t. feature ordering, agnostic w.r.t. the size and dimension $d$ of the considered distribution sample (with $d$ less than some upper bound $D$). This architecture enjoys universal approximation and robustness properties, generalizing former results obtained for point clouds (Maron et al., 2020). The merits of DIDA are demonstrated on two tasks defined at the dataset level: patch identification and performance model learning, comparatively to the state of the art (Maron et al., 2020; Jomaa et al., 2019; Muñoz et al., 2018). The ability to accurately describe a dataset in the landscape defined by ML algorithms opens new perspectives to compare datasets and algorithms, e.g. for domain adaptation (Ben-David et al., 2007; 2010) and meta-learning (Finn et al., 2018; Yoon et al., 2018), in light of kernel methods.

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
