# OpenReview forum: "Distribution-Based Invariant Deep Networks for Learning Meta-Features"
_ICLR.cc/2021/Conference — Reject_

### Official Review · AnonReviewer1 · 2020-10-15
**Interesting extension of previous work, but the experiments could be more robust to show the claimed invariances**

**Rating:** 6
**Confidence:** 4

**Review:**

The method introduces the DIDA architecture to learn from distributions and be invariant to feature ordering and size.  The authors extend the ideas proposed by Maron et al. (2020) to the continuous domain and generalize their results.  The experiments are done on two tasks.  The patch identification (out-of-distribution test) clearly show the invariance to feature and dataset size. Nevertheless, it is not clear whether the method is invariant to feature permutation.  The performance model task shows properties of the architecture to predict global structures of the dataset within their meta-features.

Pros:
+ The paper is well written and easy to follow (there are some minor errors or descriptions that need to be improved, but they are not major issues).
+ The extension of Maron et al. (2020) is interesting and provides theoretical arguments.

Cons:
- On the other hand, the experimental section is weak.  Since the main claim is the invariant meta-features (and architecture) to permutations, I would expect to see some experiments showing it.  I see the theoretical guarantees given, but an experiment showing the improvements and benefits will make the results more robust.
- The experimental protocols need to be clearly defined.  If space is an issue, I recommend using the appendix for the details.  Despite that, authors manage to show improvement over Dataset2Vec and DSS methods.  In their current form, I found it will be difficult to reproduce the experiments.

Due to the problems on the experimental details, and simple tests I'm giving the paper a score 5.

Comments:

- I found the definition of $\sigma_X (x)$ rather strange since it depends on its inverse (i.e., it is circular).  Are you just saying that $\sigma_X$ is a permutation feature-wise of x?
- I found the bracket notation rather misleading in (1).  It seems that you are concatenating the output of $f_\varphi$ instead of creating a set for the discrete distribution $\mathbf{Z}_n$ ($\mathbb{R}^r$).
- It is not clear the shape of the matrix $A_u$.  Assuming the concatenation of the slices of $x$ to be a $2 \times 1$ column vector, and $b_u$ a column vector $t \times 1$, then $A_u$ should be the transposed of what is stated, i.e., $t \times 2$.  Review your notation convention and be clear on what type of vector column/row you are using, and how the dimensions of your matrices are given.

- What is the exact training protocol for the samples in the given datasets when training to extract the meta-features?
- Are you applying a particular set of permutations $S_{d_X}$ to the inputs (or features) while training?
- Are the DIDA networks shared for the patch identification task?  Are you training them simultaneously with the "meta cross entropy loss" (5)?  In the case of the patches, how are they selected?  It is not clear what you mean by "retaining samples with index $I \in [n]$".  Are you doing something else than extracting a subset of the features in the original sample?  I can visualize a patch in an image or 2D data, but if the data is in $\mathbb{R}^{d_X}$ is not straightforward.  Hence, a proper protocol for the extraction is needed.

- Your evaluations don't mention if you test for the performance given permutations at testing time.  Are you evaluating your method with different permutations at testing time?  What is the performance in this case?  It will be interesting to see the actual invariance working experimentally.

- In the performance modeling task, it seems from the description that the classifiers are trained directly on the dataset $\mathbf{z}$.  However, the bottom right of Fig. 1 seems to show that the classifiers are trained on the DIDA output, i.e., the meta features.  Which one is it?  Make them consistent.


Minor comments:
- Your abstract shouldn't contain citations, since most of the time it will be read outside of the paper, where the citations are missing.
- Use appropriate textual and parenthetical citations.  E.g., "by (Qi et al., 2017; Zaheer et al., 2017)" should be textual instead of parenthetical.
- Typo P2 p2: "characteristcs"

---

> ### Author Response · Authors · 2020-11-18
> **Answer to reviewer 1**
>
> “It is not clear whether the method is invariant to feature permutation”
>
> Following your remark, this was clarified in the paper (section 2.2):
> Eq. 1 details how one node in an invariant layer maps discrete distribution {z_i}, i in [[1;n]] onto discrete distribution {f(z_i)} with
> f(z_i) = 1/n \sum_j \phi (z_j,z_i),
> Where phi is defined (Eq 2; with z_i = (x_i,y_i), a single label for simplicity) as:
>     \phi(z_i,z_j) =  = v (\sum_k  u(x_i[k], x_j[k], y_i, y_j))
> with u, v are 1-layer neural networks and k varying among the features
> The invariance thus follows from the definition.
>
> In practice, this invariance is an important property, as demonstrated by a lesion study (Table 1: the performance drops from ca 81% with DIDA (1 permutation invariant layer, 323,000 parameters) to ca 65% with No-FInv-DSS, 1,300,000 parameters).
>
> The experimental setting was clarified following your recommendation (section 4 and Appendix D):
> all considered algorithms (Dida and baselines) share the same experimental setting, with same hyperparameters for the two tasks (Appendix D.3);
> The protocol for defining the patches is detailed (Appendix D.1, Table 4);
> Training procedures are given in Algorithm 1 (for Task 2) and in Algorithm 2 (Appendix D.2 for Task 1)
> The source code is in supplementary material for a full reproducibility (of Dida and the considered baselines).
>
> Are the DIDA networks shared for the patch identification task? Yes, Task 1 is achieved along a Siamese learning procedure: there are two copies of the same DIDA network, each copy operates on one patch, and the network is updated using the cross-entropy loss. All other baselines are trained in the same way.
>
> Task 2 (performance modeling) aims to rank hyper-parameters based on the performance obtained on a patch z; in order to do so, the first (DIDA or baseline) module learns meta-features characterizing z. The pseudo-code has been added in Algorithm 1.
>
> Comments:
> You are right, sigma(x) = (x_{\sigma^{-1}(1)},... x_{\sigma^{-1}(d)} corresponds to a feature-wise permutation of x. This was simplified.
>
> The bracket notation has been removed; we wanted to explicit that the matrix z  formed by the set of z_i is mapped onto the matrix made of the set of f(z_i),
> with  f(z_i) = 1/n \sum_j \phi (z_j,z_i).
>
> You are right, A_u is a (t,2) matrix that is applied to the 2-dimensional vector (x[k],x'[k]). Thanks for noting the typo.
>
> “Are you applying a particular set of permutations on features” / “Your evaluations don't mention ...”
> Please see above: f(z_i) is by construction invariant w.r.t. permutation of the features: there is no need to take care of permutations by sampling or in any other way.
>
> Minor comments:
> Thanks: they are taken into account in the revised version.

---

> > ### Comment · AnonReviewer1 · 2020-11-19
> > **Thanks for the clarification and the added experiments**
> >
> > The answers from the authors cleared my questions regarding the method.  The added experiments shed more light on the proposal performance.  I understand that with the limited time it is not feasible to add a bast amount of experiments.  Nevertheless, I would advise to do a more thorough evaluation to fully evaluate the robustness to the permutations. Not only checking for the additional layers, but also evaluating the permutations on the data itself both in training and testing.  Theoretically the building functions are invariant, but I'm concerned on which patterns the neural networks that implement them could pick up and rely on.
> >
> > I found that the paper improved, and as such I will increase my evaluation.

---

> > > ### Author Response · Authors · 2020-11-25
> > > **Additional experiments**
> > >
> > > Thank you for your encouraging words !
> > >
> > > We tried to follow your suggestions, and we investigated further the robustness to permutations along three settings:
> > >
> > > * [A] for a fixed set of features, 128 patches are extracted and their meta-feature vectors are computed (with DIDA trained on Task 1). The reference vector is the average of these meta-feature vectors.
> > > The robustness/stability is assessed from the mean and standard deviation of the distances between the meta-feature vectors and the reference vector.
> > >
> > > * [B] The 128 patches above undergo feature permutation (one feature permutation for each patch), and the associated meta-feature vectors are computed. The distances of these vectors with the reference vector defined in [A] are computed, and the mean and standard deviation of these distances are reported: they show the additional impact of the feature permutation w.r.t. setting [A].
> > >
> > > * [C] 128 patches are sampled (uniform selection of the samples and the features) and their meta-feature vectors are computed.
> > > The reference vector is the average of these meta-features. The distances of these meta-feature vectors with the (new) reference vector are computed. The mean and standard deviation of these distances show the impact of both sampling the examples and the features.
> > >
> > > The comparative results of DIDA and the baseline  No-FInv-DSS, both trained on Task 1, are reported in Appendix D.5, Figure 4, on two datasets. They show that:
> > > * for DIDA, the settings [A] and [B] yield same distributions, while a slightly higher mean and variance are obtained for setting [C]; in other words, DIDA is unaffected by feature permutation.
> > > * for No-FInv-DSS, the settings [B] and [C] yield similar distributions, suggesting that No-FInv-DSS makes no difference between permuting features and sampling new features.

---

### Official Review · AnonReviewer2 · 2020-10-27
**Overall a reasonable paper**

**Rating:** 6
**Confidence:** 3

**Review:**

##########################################################################

Summary:

The paper proposes a new deep learning architecture that is invariant under permutations of both the data points and the features. The paper shows that this new architecture also has the universal approximation property. Empirical experiments were performed to demonstrate the effectiveness of this new architecture.

Overall, the paper seems to be well-motivated. It is also technically sound and the presentation of the idea is clear. I have some comments that are detailed below.

##########################################################################

Comments:

- One of the main features of the proposed architecture is that it is invariant under the permutation of the data features. While it is intuitive that the results should be invariant under the permutation of samples, I am not fully convinced why it is desirable for the architecture to be invariant under the permutation of features. Are there some solid motivating use cases?

- It seems that the proposed architecture is related to kernel methods. Particularly, the interaction functional \phi is similar to a kernel function. The authors should discuss the connection between their method and kernel methods.



- In Section 2.3, the proposed architecture uses one invariant layer. However, in the experiments, two invariant layers were used. Why do we need one more invariant layer in the experiments?


##########################################################################

Minor comments:

- Line 6 in abstract: avoid using abbreviations in the abstract ('w.r.t.' --> 'with respect to')

- Section 4 1st paragraph last line: "allocated ca the" --> "allocated the"

---

> ### Author Response · Authors · 2020-11-18
> **Answer to reviewer 2**
>
> The reason why it might be desirable to be invariant under the permutation of features is twofold:
> * A solid motivating use case is Task 2: learning a performance model. The performance of a supervised ML algorithm on a dataset is (or should be) invariant under the permutation of the descriptive features. Learning meta-features that satisfy this invariance thus yield performance models with fewer parameters, expectedly enforcing a better generalization.
> * This claim is backed up by a lesion study: architectures that do not enforce the invariance wrt feature permutation suffer a loss of performance (Table 1).
>
>
> You are right, the proposed architecture (Eq.1) is indeed related to kernel methods: both a kernel approach and DIDA map the initial data matrix z = (z_i) onto another matrix made of the (f(z_i)).
> * However, f(z_i) is an r-dimensional vector, averaging all r-dimensional vectors phi(x_i,x_j), and thus the information contained in a particular pair of examples is lost;
> * Secondly, the "kernel" phi is learned; thirdly, it takes into account the label.
> * Overall, analyzing the relationship of distributional NNs and kernel methods seems to be very interesting (e.g. inspecting the properties of the (f(z_i)) matrix), and we mentioned it in conclusion as a perspective for further work.
>
> Concerning the number of invariant layers: in general (on Task 1, and for the performance model with k-NN), better results are obtained with 2 invariant layers instead of 1 (complementary results are added in Tables 1 and 2).
>
> Minor comments:
> Thanks: they are taken into account in the revised version.

---

### Official Review · AnonReviewer3 · 2020-10-29
**Empirical evaluation could be improved**

**Rating:** 5
**Confidence:** 2

**Review:**

The paper presents a neural network layer designed to process distribution samples that is invariant to permutations of the samples and the features. The proposed method is compared empirically to DSS, which achieves the same types of invariance but is restricted to point sets rather than discrete or continuous probability distributions. The two tasks used for the empirical evaluation in the paper are: a) patch identification (are two blocks of data extracted from the same original dataset?) and b) model configuration assessment (is one configuration of a learning algorithm going to produce a more accurate model for a particular dataset than another one?). On the first task, the paper compares to models built using Dataset2Vec embeddings as well as DSS. On the second task, the paper compares to handcrafted features as well as DSS. In both tasks, the proposed method produces more accurate predictors than DSS, etc. The paper also has some theoretical results regarding the universality of the proposed architecture and its robustness w.r.t. Lipschitz-bounded transformations.

A weakness of the paper is that its primary theoretical advantage over DSS (according to my limited understanding) is that it is applicable to distributions rather than just point clouds, but the experiments only consider point clouds (presumably to be able to compare to DSS). The main selling point is then that the proposed method outperforms DSS in this setting in the experiments. However, why this is the case is not clearly explained in the paper. The paper itself states that DSS is a special case of the proposed approach. If possible, an ablation study showing which particular aspects of the new method contribute the most to the observed difference in predictive performance seems appropriate.

Page 3 refers to "Remark 5", but this remark does not appear to exist.

Rather than using the handcrafted features directly, it would be useful to train an embedding network (e.g., a Siamese network), to yield better embeddings for the two tasks concerned. This would be an obvious approach from a practical point of view that is not considered in the paper.

Why are handcrafted features not included in the first task (Table 1)?

Why is Dataset2Vec not included in the second task (Table 2)?

The Dataset2Vec results in Table 1 are from (Jomaa et al., 2019) but Appendix D.2 states that the publicly available implementation of Dataset2Vec was used. For what? Also, are the results shown in the table for DSS, etc., obtained under exactly the same experimental conditions as those used in (Jomaa et al., 2019)? This is important to enable a fair comparison.

---

> ### Author Response · Authors · 2020-11-18
> **Answer to reviewer 3**
>
> Point clouds vs distributions: You are right, we focus on point clouds to be able to compare DIDA to existing baselines: to the best of our knowledge, there is no related prior work on distributional neural networks, considering e.g., weighted point clouds.
>
> A tentative interpretation for the fact that DIDA improves on DSS is the structure of the NN, accounting for pair interactions (section 4.1).
> An ablation study has been conducted to show the importance of feature permutation invariance (Tables 1 and 2).
>
> "Rather than using the hand-crafted features directly": indeed, we use an embedding on the top of the hand-crafted meta-features, one distinct embedding for each task. The embedding is trained using a Siamese architecture for Task 1; it is trained as a usual neural module for Task 2.
>
> Regarding the baselines, following your suggestions, we added:
> * hand-crafted meta-features (followed by a trained embedding) for Task 1;
> * Dataset2Vec (re-implemented) for Task 1 and Task 2, thus supporting the fair comparison of the algorithms under the same experimental setting.
>
> Hyperparameters of the neuronal architectures, fixed for the two tasks, are detailed in Appendix D.3 (for baselines) and Section 4 (for Dida).

---

### Official Review · AnonReviewer4 · 2020-10-31
**Official Blind Review #4**

**Rating:** 7
**Confidence:** 3

**Review:**

This paper proposes a novel set/distribution representation architecture DIDA, which leverages pairwise embedding of the set’s elements. The method can be used to represent discrete and continuous distribution representation. The authors also provide the theoretical proofs of the universality of the invariant layers, the local consistency. The experiments show that the architecture improves some dataset representation tasks

On quality: A single idea was developed and well-executed in this work. The theoretical considerations are on point and improve the understanding of the applicability of the architecture.

On clarity: This idea is rather clear but the writing and the structure of the paper are sometimes difficult to follow. For example, without a previous background in the field, it is quite hard to understand why some theoretical considerations were made. It is also difficult to understand/verify some parts of the proofs that refer to the appendix, which is missing.

On originality: Previous works on distributions representations, only consider quantities that are related to their first moments. Using pairwise interactions mainly leads to considering second moments when representing a distribution. In that sense, the paper brings some originality to the field. By also making small adjustments in their theoretical analysis, the authors make the method general enough to be applied to distributions instead of points of clouds.

Questions/Concerns:
Building the representation of pairwise interactions alone can lead to representations that ignore the first moments of the distribution. To fully characterize a distribution, all the moments should be considered (or at least the first and the second moments in your case).
Clearly, the complexity here is O(N^2), so how do you scale your method to large sets/ distributions?
Why didn’t try classical point cloud benchmarks?

---

> ### Author Response · Authors · 2020-11-18
> **Answer to reviewer 4**
>
> We thank you for your kind words.
>
> Concerning the first and second moments:
> As you noted, the pairwise interactions yield the second moments; the first moment is not accounted for as the data is normalized in a pre-processing step.
>
> The scalability of the approach is obtained by restricting the considered pairs (z_i,z_j) (Eq. 1) and requiring z_i and z_j to be in the neighborhood of each other. This could be achieved using fast neighbor search methods, e.g. quad-trees (this is for further work).
>
> Experimentally, the size of the model is ca the same for DIDA and for DSS (Table 1). The training time, under the same experimental setting for Task 1 (UCI), is ~ 1hr for DSS and ~4hr for DIDA on  NVIDIA-Tesla-V100-SXM2 GPU with 32GB.
>
> We did not use point cloud benchmarks as our goal was to investigate the (more difficult) case where the data are labelled, and see whether the automatic building of meta-features was feasible.
>
> However, DIDA can be applied to point cloud benchmarks: the definition of invariant layers (Eq. 1) covers the case where the group of invariances reflects the domain properties (e.g. invariance under rotation, translation, or permutation).

---

### Author Response · Authors · 2020-11-18
**Answer to reviewer comments**

We would like to thank all four reviewers for their time and thoughtful reviews, and for your kind words, acknowledging the “originality” of our approach, with both “on point” theoretical arguments as well as its empirical “effectiveness”.

Following your general constructive critiques, we revised the paper (additions in blue in the paper for easy checking):

I. The experimental setting is extensively detailed. We add pseudo codes for training Task 2 in the main paper (and in supplementary for Task 1). DIDA and all baseline codes are available in supplementary material;

II.  The baselines are extended to include:
* Task 1: Hand-crafted metafeatures, and reimplementation of Dataset2Vec
* Task 2: Dataset2Vec re-implemented

III.  Lesion studies have been conducted:
* Impact of removing feature-invariance (Task 1)
* Impact of 1 vs 2 invariant layers

Please find below our detailed answers to your specific comments in each review.

---

### Author Response · Authors · 2020-11-25
**New version submitted**

Dear reviewers, we have submitted a new version of our work, with added experiments on the stability of the meta-features with respect to permutations and sampling strategies (Appendix D.5, see detailed answer to #AnonReviewer1). We thank all reviewers for their constructive feedback, which has helped us improve the paper.

---

### Decision · Program_Chairs · 2021-01-07
**Final Decision**

**Decision:**

Reject

**Comment:**

This paper invariantizes distribution based deep networks by using pairwise embedding of the set’s elements.  The idea is inspired from De Bie et al. (2019), which allows invariance to be incorporated through the interaction functional.  Although the paper is well executed with solid theoretical analysis and solid response to the reviewers' comments, the novelty is limited, and reviewers have concerns with experiments and presentation.